# STEALING AND DEFENDING THE ENDS OF LLMS

## ABSTRACT

Soft prompt tuning has emerged as a powerful and automated approach for adapting large language models (LLMs) to new tasks, eliminating the need for manual prompt engineering. The practical relevance of soft prompts is underscored by their support in major toolkits and APIs such as NVIDIA NeMo and IBM Watsonx AI. However, as soft prompts encode valuable, task-specific information, they have become attractive targets for adversarial extraction. In this work, we demonstrate that attackers can extract functionally equivalent soft prompts from prompt-tuned LLMs, effectively replicating their capabilities without access to the original training data or resources. By training a dedicated inversion model, we show that such extraction generalizes, enabling recovery of soft prompts for any downstream task on the given model. To counter this threat, we introduce CAP (**C**overage-**A**ware **P**erturbation), an active defense that substantially impairs extraction while maintaining task performance for legitimate use. Our framework highlights both new risks and practical solutions, paving the way for more trustworthy deployment of adapted LLMs.

## 1 INTRODUCTION

Large language models (LLMs) exhibit strong in-context learning capabilities (Brown et al., 2020; Radford et al.), enabling them to perform a wide range of downstream tasks simply by prepending an appropriate prompt to the input (Gao et al., 2021; Raffel et al., 2020; Shin et al., 2020), without modifying the LLM parameters. Building on this idea, soft prompt tuning (Lester et al., 2021; Li & Liang, 2021; Liu et al., 2022b) has emerged as a powerful adaptation technique. Rather than updating the full model, it tunes a small set of additional learnable parameters added within the model's input embedding space. This approach is an instance of parameter-efficient fine-tuning (PEFT) (Lester et al., 2021) methods, a set of LLM adaptation techniques that adapt a pre-trained LLM by updating a small fraction of parameters. In the case of prompt tuning, PEFT is implemented by adding trainable embeddings, which are optimized via backpropagation, while other weights in the LLM are frozen. At inference time, these learned soft prompts are loaded and injected into the input embedding layer, enabling the LLM to generate task-specific predictions.

Platforms like NVIDIA's NeMo (Harper et al., 2024) and IBM Watsonx AI (IBM, 2025) enable prompt tuning in practice. They help practitioners to optimize soft prompts to cater to downstream tasks while also allowing deployment of these prompt-tuned models. Once deployed, these models can be queried via hosted APIs that return model predictions. While these platforms facilitate a black-box query setup, they raise multiple privacy risks regarding the adaptation data (Bailey et al., 2023; Duan et al., 2023; Hanke et al., 2024; Lester et al., 2021) and undermine the intellectual property of the party that trained the prompts (Maini et al., 2024; Freethink, 2025). Prior work (Wang et al., 2025) has shown that prompt-tuned models are vulnerable to membership inference attacks. Besides that, adversaries can leverage query-access to a prompt-tuned LLM to extract the learned soft prompt to host their own copy of the prompt-tuned model, aiming to replicate the victim model's downstream performance. As real-world APIs, such as NVIDIA NeMo and IBM Watsonx AI, offer the possibility of exposing prompt-tuned LLMs, this threat becomes real. Prior works leverage the intuition that LLM outputs, such as next-token probability vectors, encode significant residual information about the preceding input and thereby focus on inverting next-token probability vectors to extract discrete textual prompts (Morris et al., 2024). However, inversion attacks, in the context of soft prompts, remain unexplored.

We introduce a novel two-staged soft prompt extraction and inversion attack against prompt-tuned LLMs. This attack includes two stages, distillation and inversion. Our attack first follows a distillation-based approach to reconstruct a functionally equivalent version of the target LLM's tuned prompt and further aims to extract soft prompts across multiple downstream tasks by inverting next-token probability vectors. In distillation, the first stage of the attack, we aim to reconstruct a *behavioral clone* of the target LLM's tuned soft prompt on a specific downstream task using black-box query-access to the prompt-tuned LLM. The experimental results demonstrate how an adversary can successfully reconstruct a target LLM's tuned prompt by optimizing a randomly initialized prompt embedding, such that it mirrors the target prompt-tuned model's output probability distribution. Notably, our attack is successful even when the adversary relies on out-of-distribution (OOD) queries, underscoring the robustness of our attack. During inversion, the second stage, the attacker extends the initial distillation approach by training a model that inverts a prompt-tuned LLM's outputs into soft-prompt embeddings. The resulting inversion model is optimized to generalize across tasks, enabling the extraction of *any* soft prompt tuned for this model, even for previously unseen downstream tasks without additional training.

Given the severity of the threat, we introduce an active defense, *Coverage-Aware Perturbation (CAP)*, against soft prompt extraction. Our defense leverages the insights that, to invert from probability vectors to soft prompts with higher performance, an adversary needs to query the prompted models with highly diverse queries. This stands in contrast to benign users who usually query for one or a few concrete downstream tasks (Dubinski et al., 2023). We notice that with the increasing query diversity, the query latent (embedding) space coverage increases too, enabling the detection of extraction attempts by monitoring this coverage. After every user-query, CAP estimates the coverage and penalizes adversaries based on the estimation. The adversaries are penalized by perturbing the tuned soft prompt embeddings to thwart the soft prompt extraction attacks that target the input end of the LLM and by perturbing the model responses to defend against the last-layer extraction attacks that target the output end. CAP not only prevents soft prompt extraction attacks by adversaries but also maintains performance for benign users. Thus, CAP effectively protects both ends of the LLM, preventing extraction at the input (soft prompts) and output (last-layer weight matrix).

Our experimental evaluation across various natural language processing (NLP) tasks shows that the soft prompts inverted from the probability vectors achieve downstream task performance comparable to that of the target prompt-tuned model. Finally, our attack is significantly more efficient than tuning a soft prompt from scratch. Although our soft prompt extraction attack is robust, our experimental results show that CAP defense successfully prevents it. Beyond protecting against soft prompt extraction, we also show that CAP is able to protect against state-of-the-art last-layer extraction attacks (Carlini et al., 2024). To provide a concrete example, the root mean square error (RMSE) between the original and the extracted weight matrix of the final layer in the T5-base model (Raffel et al., 2020) increases from a negligible 1.96e-5 to a substantial 18.21 when our defense is applied. Thus, CAP is able to protect prompt-tuned LLM APIs against the stealing of both their ends.

In summary, we make the following contributions:

- We propose a **novel two-staged black box prompt extraction and inversion attack** that enables an adversary to invert prompt-tuned LLM outputs and extract functionally equivalent soft prompts across multiple downstream tasks even with OOD queries.

- To mitigate this threat, we propose **CAP, an active defense to prevent the extraction of both the ends of LLM APIs** by monitoring the adversaries' query diversity and accordingly penalizing them by perturbing the fixed query-invariant soft prompt to defend against inversion attacks on the input end of the LLM and perturbing model outputs to defend against the last-layer extraction attacks on the output end of the LLM.

- Through our **thorough experimental evaluation**, we demonstrate that soft prompts inverted from probability vectors provide downstream performance comparable to their original counterparts on text classification and natural language inference tasks. CAP maintains high performance for benign users while successfully protecting against inversion and last-layer extraction attacks on prompt-tuned LLMs.

## 2 BACKGROUND AND RELATED WORK

We provide an overview of adaptation techniques for LLMs with a focus on soft prompts, followed by an analysis of model inversion and extraction attacks, along with the corresponding state-of-the-art defense mechanisms. Additional background can be found in Appx. A.

**LLM Adaptations with Soft Prompts.** LLMs can be adapted to downstream tasks by (1) *adapting their inputs* using discrete textual prompts (Brown et al., 2020; Gao et al., 2021) and continuous parameters with either soft prompts (Lester et al., 2021; Liu et al., 2022b) or prefix tuning (Li & Liang, 2021); (2) *adapting the internal layers* with methods like low-rank adaptations (e.g., LoRA (Hu et al.) or AdaLoRA (Zhang et al., 2023), and most of other PEFT (Parameter Efficient Fine Tuning) methods (Han et al., 2024; Liu et al., 2022a), which add additional parameters (usually a small number) within the model, and (3) *full or last layer fine-tuning* (Gao et al., 2021; Raffel et al., 2020). The input-based adaptations based on prompting gained substantial popularity since they achieve high performance and do not require keeping separate model parameters per downstream task for inference (Lester et al., 2021; Li & Liang, 2021), in contrast to the full fine-tuning or other PEFT methods. Thus, we turn our attention to prompts. Discrete prompts require prepending the input queries with textual instructions and demonstrations (also referred to as shots) to solve a given downstream task (Gao et al., 2021). The main drawback is the requirement to find such prompts in the discrete space (Shin et al., 2020). To eliminate the obstacle, soft prompts add additional trainable parameters in the input embedding layers of LLMs (Lester et al., 2021; Liu et al., 2022b; 2024) —enabling standard backpropagation to the soft prompt parameters using (usually private) data for downstream tasks (Duan et al., 2023; Hanke et al., 2024). Prefix tuning is a very similar approach to soft prompts, but apart from the input embeddings, it also adds additional parameters as inputs to each (attention) layer of an LLM (Li & Liang, 2021). We focus on soft prompts, which have not been explored yet for model inversion attacks.

**Inversion Attacks in Vision and Language Models.** Several successful inversion attacks in image and natural language processing domains demonstrated that approximate reconstruction of inputs can be achieved, given logits or probability outputs. (Fredrikson et al., 2015) were the first to show that machine learning models can leak identifiable and sensitive information about their training data, such as *users' faces* or *genotypes* (Fredrikson et al., 2014), even when accessed as black boxes since the model outputs (especially softmax outputs) can be exploited for reconstruction attacks. (Teterwak et al., 2021) also demonstrated that a surprisingly high amount of information about input images can be approximately reconstructed from the logits of a discriminatively trained classifier. In the language domain, successful recovery of the input text sequence was achieved from text embeddings by conditioning the encoder from an encoder-decoder transformer model as a part of the inversion process (Li et al., 2023; Morris et al., 2023). Additionally, (Morris et al., 2024) succeeds in performing inversion from probability distribution to discrete (textual) prompts by recovering text input from probability outputs of language models.

**Model Extraction Attacks.** Black-box access to the model enables not only the reconstruction of its training inputs but also the recovery of the model itself (Dziedzic et al., 2022a; Jagielski et al., 2020; Tramèr et al., 2016). Language model extraction has become challenging due to the secrecy of details regarding the model size, architecture, datasets, and training process (Achiam et al., 2023). However, there are still many attempts to extract isolated components of language models, namely decoding algorithm (Naseh et al., 2023), the model's embedding size (Carlini et al., 2024; Finlayson et al.), sentence encoders (Dziedzic et al., a), functionality of the last fine-tuning layer (Krishna et al.), and, the most prominent, the weight matrix of the last layer (Carlini et al., 2024). This latest attack leverages the observation that the final layer of many LLMs behind APIs performs a projection from the hidden representation to a higher-dimensional logit vector. Thus, the final layer is low-rank, and sending random queries and observing when they become linearly dependent indicates the dimension of the hidden representations. The attack can be further extended to recover the final output projection matrix that maps from the final hidden layer to the output logits.

**Defenses against Model Extraction.** Defenses against model extraction can be primarily categorized into three types, following (Dziedzic et al., b), namely active defenses that act while extraction is happening and, e.g., perturb the responses to poison the training objective of an attacker (Dubinski

et al., 2023; Mazeika et al., 2022; Orekondy et al., 2020; Wu et al., 2024), passive defenses analyze the distribution of incoming queries and stop answering if they detect an outlier set of queries (Kesarwani et al., 2018; Juuti et al., 2019; Chen et al., 2020), and reactive defenses, also known as post-hoc verifications, which try to prove a model theft rather than preventing the attack from happening (Adi et al., 2018; Dziedzic et al., 2022b; Jia et al., 2021). The LLM APIs are designed to be highly responsive and the interruption of service is not acceptable, thus eliminating the passive defenses. The ends of LLM APIs, such as soft prompts or the last layer, are composed of a relatively small number of parameters, thus lowering the effectiveness of watermarking-based reactive defenses. Active defenses are highly desired in the LLM API setting to defend against extraction as it is happening. Therefore, in this work, we build on the latest type of active defenses (Dubinski et al., 2023; Dziedzic et al., b) to estimate the information leakage incurred by the queries to the LLM API and then perturb the high-dimensional outputs according to this estimated leakage (Dubinski et al., 2023).

# 3 TWO-STAGED BLACK-BOX SOFT PROMPT EXTRACTION AND INVERSION ATTACK

**Setup and Threat Model.** We consider an LLM provider who deploys a prompt-tuned LLM, and a user. We adopt the scenario, well-suited to real-world deployment settings where a user has black-box API query access to the prompt-tuned LLM and obtains a next-token probability distribution across the vocabulary. Our experimental evaluation is based on varying levels of probability access, as many API services expose only top-k probabilities in practical scenarios (Achiam et al., 2023; Cohere, 2025; Anil et al., 2023). Our two-staged attack does not assume access to the underlying model architecture or training data from the prompt-tuned LLM.

**Problem.** We consider the problem of inverting the next-token probability vectors (prompt-tuned model's outputs) to extract functionally equivalent soft prompts that replicate a downstream performance on the attacker's model that employs the extracted prompt, comparable to the target prompt-tuned model across multiple downstream tasks. To achieve this, we present a detailed overview of both the stages of our attack, as illustrated in Figure 1.

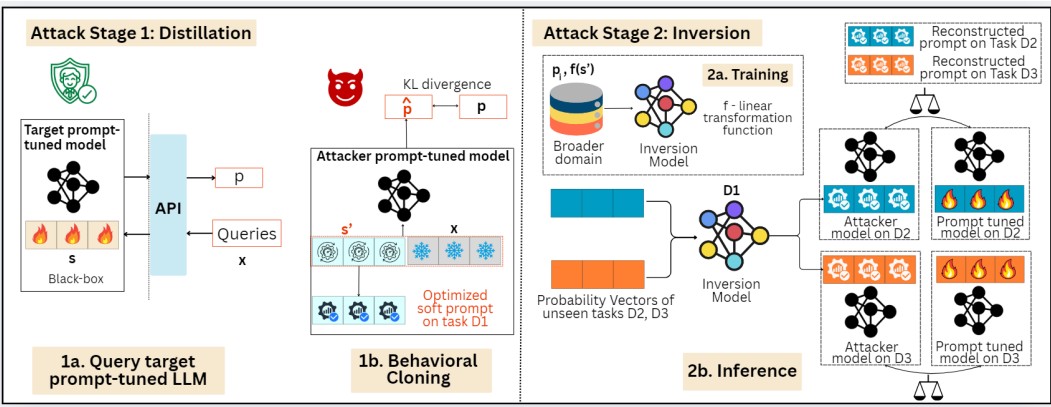

Figure 1: **Attack Stage 1: Functional prompt extraction attack using distillation approach.** The adversary, with black-box access to the prompt-tuned model, (1a) queries it with text inputs and collects output distributions. (1b) A randomly initialized soft prompt is then optimized to minimize the KL divergence between the surrogate model's predictions and the victim's outputs, yielding a functionally equivalent prompt. This approach replicates the victim's behavior on a given task without recovering the exact tuned soft prompt. The downstream performance of the target prompt-tuned model is compared to that of the attacker's model to evaluate the success of this extraction attack. **Attack Stage 2: Inversion across multiple downstream tasks.** Leveraging access to the functionally equivalent soft prompt from stage 1, (2a) the adversary trains an inversion model on LLM outputs and corresponding extracted soft prompt (task D1), such that (2b) the inversion model generalizes on unseen tasks, and produces tuned soft prompts for D2 and D3. The downstream performances on the attacker's prompt-tuned models are further compared to the models tuned on tasks D2 and D3, to evaluate the success of the inversion attack.

### 3.1 STAGE 1: DISTILLATION

This stage aims to reconstruct a functionally effective soft prompt just from black-box query access to the prompt-tuned LLM, which uses prompt $\mathbf{s} \in \mathbb{R}^{T \times d}$, where $d$ is the model's embedding dimension. T is the number of virtual tokens/soft prompt length. As illustrated in Figure 1, the adversary queries the prompt-tuned LLM via API with $N$ text inputs $\{x_i\}_{i=1}^N$, either in-distribution (ID) or out-of-distribution (OOD). To extract functionally equivalent soft prompts that will help yield performance that matches the prompt-tuned LLM, the adversary queries *diverse* $N$ text inputs, following the intuition in (Zhao et al., 2025) and obtains the probability distribution $\{p_i\}_{i=1}^N$ over the entire vocabulary, where each $p_i \in \mathbb{R}^{|\mathcal{V}|}$ is a probability vector over the vocabulary $\mathcal{V}$. The probability vectors are then mapped to the corresponding text inputs to form $(x_i, p_i)$ pairs. The adversary further considers a pre-trained LLM of preferably the same architecture as the target prompt-tuned model. The text inputs $\{x_i\}_{i=1}^N$ are queried to an LLM that initially employs a randomly initialized soft prompt $\mathbf{s}' \in \mathbb{R}^{T \times d}$. Thus, the adversary prepends embeddings of $\mathbf{s}'$ to the text input embeddings of $x_i$, and further optimizes only $\mathbf{s}'$, such that the attacker model's outputs $\{\hat{p}_i\}_{i=1}^N$ using the optimized soft prompt mirror the outputs of the prompt-tuned LLM, $\{\mathbf{p}_i\}_{i=1}^N$. The randomly initialized soft prompt is optimized by minimizing the Kullback–Leibler (KL) divergence (Kullback & Leibler, 1951) between the attacker model and the target prompt-tuned model's probability distributions. Formally, the optimization problem is:

$$\mathbf{s}' = \arg\min_{\mathbf{s}} \frac{1}{N} \sum_{i=1}^N \mathrm{KL}\big(\mathbf{p}_i \parallel \hat{\mathbf{p}}_i\big), \tag{1}$$

where $\mathbf{p}_i$ is the probability distribution of the target prompt-tuned LLM for input $x_i$ and tuned prompt $\mathbf{s}$, and $\hat{\mathbf{p}}_i$ is the probability distribution of the attacker model with learnable randomly initialized soft prompt $\mathbf{s}'$ for the same input $x_i$. The downstream performance is computed for both the prompt-tuned model (that uses $\mathbf{s}$) and the attacker model (that uses $\mathbf{s}'$). If the performances are comparable, the attack is said to be successful based on the key insight that soft prompts that share similar functional capabilities steer the LLM to produce comparable downstream performances on a specific downstream task.

### 3.2 STAGE 2: INVERSION

While stage 1 reconstructs functionally equivalent task-specific prompts, extending it to multiple downstream tasks is computationally expensive and undermines the very purpose of stealing functionally effective soft prompts. Thus, in this stage, an adversary utilizes $\{p_i\}_{i=1}^N$ obtained in stage 1 and maps them to the linear transformations of the $\mathbf{s}'$, a reconstructed version of the target LLM's task-specific soft prompt obtained in stage 1. We use this dataset to train an inversion model. The next token probability vectors are projected into a dimension suitable for a transformer to process. These projected probability vectors are encoded by passing them through a stack of transformer layers and decoded into a sequence of prompt embeddings. Given another prompt-tuned model's (unseen downstream task) next-token probability distribution as an input, the inversion model outputs functionally similar soft prompts across multiple downstream tasks, thereby demonstrating the inversion model's generalization ability.

## 4 EMPIRICAL EVALUATION

**Downstream Tasks.** To evaluate the effectiveness of our reconstructed soft prompts, we use standard NLP datasets, namely SST2, MNLI, and YELP from the GLUE benchmark (Wang et al.), rotten-tomatoes movie review (denoted as MOVREV), Amazon Polarity (denoted as AMAZON) datasets from the Hugging Face datasets library (Lhoest et al., 2021), and IMDB (Maas et al., 2011). These datasets correspond to classification tasks such as sentiment analysis (SST-2, IMDB, YELP, MOVREV, AMAZON) and natural language inference (MNLI). The data from these downstream datasets is used to tune the soft prompts simulating the task of model owners, thereby resulting in distinct prompt-tuned LLMs for evaluation.

Table 1: **Extraction of functionally equivalent soft prompts with ID and OOD queries in distillation attack stage.** We report downstream accuracy (%) for the target prompt-tuned LLM, the adversary's prompt-tuned LLM, the target model with randomly initialized prompts and target model without soft prompt across different tasks. Both the target and adversary's LLM use T5-base as the backbone architecture.

| Task | Query Type | Target (%) | Adversary (%) | Random (%) | Zero-shot (%) |
|---|---|---|---|---|---|
| AMAZON | ID | 90.89 | 91.08 | 49.80 | 49.80 |
| MNLI | ID | 77.51 | 75.77 | 50.77 | 50.54 |
| YELP | ID | 93.69 | 93.57 | 46.70 | 51.90 |
| MOVREV | ID | 82.27 | 81.61 | 51.80 | 46.60 |
| SST-2 | ID | 93.90 | 92.32 | 50.60 | 49.08 |
| YELP | OOD | 93.69 | 90.60 | 51.80 | 51.90 |
| AMAZON | OOD | 90.89 | 90.69 | 49.80 | 49.80 |
| MOVREV | OOD | 89.20 | 86.00 | 46.70 | 50.09 |

Table 2: **Extraction of functionally equivalent soft prompts on unseen downstream tasks in the inversion attack stage.** We report downstream accuracy (%) of the target prompt-tuned LLM (clean), the adversary's LLM using a reconstructed soft prompt (without CAP), the same adversary model when CAP is enabled, the target model with randomly initialized embeddings, and the zero-shot accuracy of the base model.

| Training Task | Evaluation Task | Target (%) | Adversary (CAP Off) (%) | Adversary (CAP On) (%) | Random (%) | Zero-shot (%) |
|---|---|---|---|---|---|---|
| YELP | AMAZON | 90.20 | 88.20 | 49.90 | 49.80 | 49.80 |
| | MOVREV | 82.10 | 81.81 | 46.70 | 46.70 | 46.60 |
| AMAZON | YELP | 88.80 | 87.20 | 51.80 | 51.80 | 51.90 |
| | MOVREV | 82.10 | 80.70 | 46.70 | 46.70 | 46.60 |

**Models.** While different downstream tasks yield different prompt-tuned LLMs, the model architecture they share is the same. We primarily consider T5-base backbone (Raffel et al., 2020), consisting of 222M parameters, as the underlying model for our two-staged attack. Additionally, we also evaluate our attack on varying model architectures T5-small, T5-large (Raffel et al., 2020) and roberta-base (Liu et al., 2019). We set the prompt length to 20 (virtual tokens for encoder and decoder) for all experiments on T5 model variants, while for roberta-base, we set it to 30. In the inversion stage of our attack, we train the inversion model using the Adam optimizer with a learning rate of $1 \times 10^{-4}$ for 8 epochs. In Appx. E, we present details about hyperparameters.

**Metrics.** To assess the success of the attack, we report the downstream performance on the prompt inverted from next-token probability vectors and compare it to that of the target prompt-tuned model. Based on the findings that multiple near-optimal soft prompts exist for a task that lie in the same low-dimensional subspace of the embedding space and that they achieve comparable performance on a downstream task (Zheng et al., 2024b), these metrics are an indicator of the reconstructed soft prompt's functional similarity to the original one.

## 4.1 RESULTS

Our experimental results are demonstrated in Table 1 and Table 2 for distillation and inversion stages of our two-staged attack on prompt-tuned LLMs.

**Distillation stage.** We evaluate the effectiveness of the distillation stage across multiple downstream tasks. Table 1 provides a comparison between the downstream performances for target prompt-tuned LLM, attacker's prompt-tuned LLM, and LLM that employs a randomly initialized soft prompt, for baseline comparison. The results indicate that by issuing in-distribution queries to the prompt-tuned

LLM, the adversary's tuned LLM achieves performance closer to that of the target LLM, across downstream tasks. For example, on SST-2 and YELP, the attacker's prompt-tuned model achieves 92.32% and 93.57% accuracy, respectively, compared to the target model's 93.90% and 93.69%. Furthermore, using diverse queries allows adversaries to capture the full functional behavior of the soft prompts even when they lack knowledge of ID queries. Therefore, results in Table 1 show that issuing OOD queries to the prompt-tuned LLM allows an adversary to yield downstream performance comparable to that of the target tuned LLM. We provide additional details about the OOD queries used in the experiments in Appx. E. Besides, with a partial (top-5 probabilities) distributional access, this attack can successfully recover the soft prompt's functionality, yielding comparable downstream performance and minimal performance degradation compared to full distributional access (see Table 9). This demonstrates that by merely by a query-access to a black-box prompt-tuned LLM for a specific downstream task, it is possible to steal the model's functionality to yield a comparable downstream performance. Even when the adversary's underlying model architecture is different from the target prompt-tuned LLM, this attack functions well (see Table 7). Additionally, our experiments on roberta-base in Table 8 indicate the success of our attack on encoder-only architectures that output a probability distribution over the classes the LLM was trained on.

**Inversion stage.** In the inversion stage, we conduct a cross-dataset evaluation. Our trained inversion model is evaluated on multiple downstream datasets, the same datasets that were used to prompt-tune a pre-trained LLM, but the inversion model has not seen the probability vectors generated by the model tuned on these tasks during training. From an adversary's standpoint, an inversion model that generalizes to unseen tasks and outputs functionally equivalent soft prompts helps yield comparable downstream performance, saving a significant amount of computational expense. We consider multiple combinations for training and evaluation datasets. Notably, the attacker's model, which leverages the reconstructed and functionally equivalent soft prompt, achieves performance comparable to that of the target prompt-tuned LLMs, thereby maintaining strong reconstruction results across unseen evaluation tasks. For instance, in Table 2, when the inversion model is trained on Amazon Polarity and generates task-specific soft prompts for YELP and MovRev, the attacker's model using these reconstructed task-specific prompts achieves a downstream accuracy of 87.20% and 80.70%, respectively, which closely matches 88.80% and 82.10%, the target prompt-tuned LLM's downstream performances for the respective tasks. Randomly initialized prompts, as expected, yield accuracy close to 50% for every task, highlighting that the attacker model's reconstructed prompts significantly surpass the performance with randomly initialized soft prompts. Overall, these results signify that our inversion method not only reconstructs high-utility functionally effective task-specific prompts but also generalizes across multiple downstream tasks.

## 5 ACTIVELY DEFENDING AGAINST INVERSION ATTACKS

Given the high vulnerability of prompt-tuned LLMs to our two-staged attack, we now turn to defenses. We introduce an active defense **Coverage-Aware Perturbation (CAP)**. Our defense strategy is coverage-aware, recognizing that our attack is successful due to the wide diversity of queries to the prompt-tuned LLM. CAP leverages the fact that adversaries assume a deployed model uses a *fixed* stealing target (tuned soft prompt) during inference. Thus, every time the adversary queries the deployed prompt-tuned LLM, the underlying soft prompt remains the same, making the LLM's behavior query-invariant. This allows our attack to recover a functionally equivalent soft prompt. Therefore, CAP continually drifts the target soft prompt (ground-truth) during inference, breaking the query-invariance and making the attack substantially hard for the adversary. We first explain how our CAP defense detects adversarial behavior and distinguishes it from a legitimate user's behavior and further describe the components of our CAP defense.

### 5.1 CAP DISTINGUISHES BETWEEN ADVERSARIAL AND LEGITIMATE QUERIES

Our setup and threat model expose a prompt-tuned LLM, which can be queried by adversarial and legitimate users alike. (Dubinski et al., 2023) show in their defense against stealing image encoders that it is possible to distinguish between adversarial and legitimate users based on the fraction of embedding space they occupy. Following the same intuition for prompt-tuned LLMs, we observe that legitimate users typically query a prompt-tuned LLM to solve a specific downstream task. However, an adversary, intending to steal the functionality of the soft prompt, queries the prompt-tuned LLM

with diverse and random inputs to capture the complete functional behavior of the soft prompt. Based on this intuition, we design our CAP defense. CAP notes that legitimate users remain task-focused when querying and gradually explore the prompt-tuned LLM's embedding space. However, adversarial users probe the LLM aggressively with highly diverse and spread-out queries that cover a major part of the embedding space. By tracking these differences in embedding space exploration, CAP successfully distinguishes between legitimate and adversarial users.

## 5.2 COMPONENTS OF CAP

**Track Embedding Space Coverage.** To effectively monitor the prompt-tuned LLM's embedding space, CAP partitions the embedding space into discrete buckets using Local Sensitive Hashing (LSH) (Indyk & Motwani, 1998). LSH enables approximate nearest neighbor (ANN) search in high-dimensional spaces by hashing similar objects in the same hash bucket based on a similarity metric. Considering the high-dimensional nature of input embeddings, CAP adapts LSH to input embeddings to measure the diversity of user queries. The overall coverage of the embedding space is computed using three metrics, *Bucket Coverage* ($C$), which quantifies the fraction of buckets occupied by the previous query embeddings, *New Bucket Rate* ($N$), which monitors the rate of newly filled buckets with increasing number of queries and *Spread* ($S$) that captures how far the embeddings are from each other. Collectively, these metrics provide a clear intuition on how much of the prompt-tuned LLM's embedding space is explored and how varied the input queries are. Further, a cost function is calibrated based on this coverage.

**Map Coverage to Perturbations.** The total cost is composed of four terms (see Equation (2)). The first term, $\lambda$, represents a minimal baseline perturbation, the second term is the coverage penalty, where $C$ represents the bucket coverage of incoming queries, $\alpha$ is a global scaling factor, $\beta$ represents the sensitivity of the coverage penalty to the bucket coverage, and $w_c$ determines the contribution of coverage penalty in the total cost. This exponential function ensures that the utility of the benign users is preserved while heavily penalizing the adversaries. The third term is the new bucket rate penalty, $\alpha\, w_n\, N$, where $N$ represents the change in the coverage. In other words, it measures how many new buckets were activated/filled with every incoming query as compared to previous queries, and $w_n$ is its weight. This term penalizes the adversaries who probe the prompt-tuned LLMs with queries that expand the overall coverage. The fourth term is the spread penalty, $\alpha\, w_s\, \min(S/S_{\max}, 1)$, where $S$ is the average distance of the embeddings from their mean and $S_{max}$ caps the penalty. The LLM provider can configure these values to control the degree of penalization.

$$\text{TotalCost} = \underbrace{\lambda}_{\text{baseline}} + \underbrace{w_c\left(\left(\frac{\alpha}{\lambda}\right)^{C/\beta} - 1\right)}_{\text{Bucket coverage penalty}} + \underbrace{\alpha\, w_n\, N}_{\text{New bucket rate penalty}} + \underbrace{\alpha\, w_s\, \min\left(\frac{S}{S_{\max}}, 1\right)}_{\text{Spread penalty}} \quad (2)$$

**Penalize adversaries.** We significantly degrade the utility on downstream tasks for an attacker extracting soft prompts using API query access by adding Gaussian noise either to the target prompt-tuned LLMs' prompt embeddings (input end of LLM) or outputs (output end of LLM) to defend against inversion and last-layer extraction attacks, respectively. The noise added is with a standard deviation $\sigma$ computed using Equation (2). We observe in Figure 4 that downstream performance decreases sharply with increasing Gaussian noise scale. This shows that the returned model outputs are less useful and become inconsistent for further training or processing, thereby successfully mitigating the extraction attacks (see Table 3 and Table 2).

## 6 EMPIRICAL EVALUATION OF OUR CAP DEFENSE

We perform experiments on a wide range of prompt-tuned LLMs with the following pre-trained models: Pythia (Biderman et al., 2023) 1.4B and 6.9B, GPT-2 (Radford et al.) Small, T5-base, and T5-small (Raffel et al., 2020) to prove the effectiveness of our defense. Similar to (Carlini et al., 2024), we use the root mean square error (RMSE) between the actual and recovered final layer weight matrix as a metric to measure the success of the extraction attack.

Our CAP defense method successfully prevents the inversion from probability vectors to soft prompts. Concretely, the accuracy of the reconstructed soft prompts inverted from the probability vectors when CAP is enabled is on par with the random performance. CAP is also effective beyond classification tasks, such as summarization (see Table 11). Furthermore, based on the experimental results in Table 4,

Table 3: **CAP defense against extraction of functionally equivalent soft prompts in the distillation stage.** CAP significantly penalizes the adversaries while maintaining high downstream performance for legitimate users. (legitimate users denoted by LEGIT, adversaries denoted by ATTACK). Both models use T5-base as the backbone.

| Task | Type of Query | #Queries | CAP Mode | Accuracy | Random | Zero-shot (%) |
|---|---|---|---|---|---|---|
| AMAZON | ATTACK | 1000 | OFF | 91.08 | | |
| | LEGIT | 1000 | ON | 86.70 | 49.80 | 49.80 |
| | ATTACK | 1000 | ON | 52.90 | | |
| YELP | ATTACK | 1000 | OFF | 93.60 | | |
| | LEGIT | 1000 | ON | 88.20 | 51.80 | 51.90 |
| | ATTACK | 1000 | ON | 51.40 | | |
| MOVREV | ATTACK | 1000 | OFF | 86.00 | | |
| | LEGIT | 1000 | ON | 78.54 | 46.70 | 46.60 |
| | ATTACK | 1000 | ON | 50.86 | | |

we find that CAP also effectively protects against last-layer extraction (see Appx. D to find details on the attack). The results show that when CAP is enabled, there is a significantly higher RMSE between the original and extracted layer, compared to when CAP is disabled—highlighting its protectiveness. The perturbation added to the logits also causes a substantial discrepancy in the extracted hidden dimensionality of the model, thereby preventing accurate dimension recovery. Based on the results in Table 3, we also show that applying CAP does not affect the model performance of legitimate users much, while degrading the performance significantly for attackers.

Table 4: **CAP against the last-layer extraction.** We report the metrics of the extraction attack and our CAP defense. We issue a different number of discrete prompt queries, measure the coverage of the occupied latent space (%) by the queries' embeddings, report the Noise Level, the size of the Hidden Dimension, and the size of the Stolen Dimension with and without our defense, similarly for RMSE between the original and extracted parameters. ***Our defense effectively prevents the stealing of the hidden dimension size and the final-layer parameters.***

| Model | Queries | Coverage (%) | Noise Level | Hidden Dim | Stolen Dim (CAP disabled) | Stolen Dim (CAP enabled) | RMSE (CAP disabled) | RMSE (CAP enabled) |
|---|---|---|---|---|---|---|---|---|
| Pythia 1.4B | 5000 | 98.97 | 9.75 | 2048 | 2048 | 4996 | $2.29 \times 10^{-7}$ | $2.07 \times 10^{-2}$ |
| | 10000 | 99.76 | 9.99 | 2048 | 2048 | 9995 | $1.73 \times 10^{-8}$ | $2.07 \times 10^{-2}$ |
| T5-small | 5000 | 62.50 | 3.02 | 512 | 512 | 1 | $6.894 \times 10^{-4}$ | $2.250 \times 10^{1}$ |
| | 10000 | 63.28 | 3.10 | 512 | 512 | 1 | $4.480093 \times 10^{-4}$ | $2.250 \times 10^{1}$ |
| T5-base | 5000 | 51.56 | 2.046 | 768 | 768 | 1 | $5.02164 \times 10^{-5}$ | $1.8211 \times 10^{1}$ |
| | 10000 | 53.12 | 2.168 | 768 | 768 | 1 | $3.06884 \times 10^{-5}$ | $1.8211 \times 10^{1}$ |
| Pythia-6.9B | 5000 | 90.6 | 7.539 | 4096 | 4096 | 4996 | $1.01 \times 10^{-6}$ | $1.82 \times 10^{-2}$ |
| | 10000 | 97.85 | 9.424 | 4096 | 4096 | 9995 | $3.298 \times 10^{-7}$ | $1.82 \times 10^{-2}$ |
| GPT-2 Small | 1000 | 63.60 | 7.07 | 768 | 762 | 448 | $7.41 \times 10^{-2}$ | $1.43 \times 10^{-1}$ |
| | 2500 | 91.70 | 19.99 | 768 | 769 | 448 | $2.2 \times 10^{-3}$ | $1.43 \times 10^{-1}$ |
| | 10000 | 100 | 26.87 | 768 | 769 | 448 | $1.3 \times 10^{-3}$ | $1.429 \times 10^{-1}$ |

# 7 CAP DEFENSE AGAINST ADAPTIVE SYBIL ADVERSARIES

We consider an adversary who queries the API from $n$ accounts. For every account, the model outputs (probabilities), which are released to the adversary. The adversary collects these model outputs to train the inversion model. Sybil adversaries try to circumvent our defense by carefully partitioning their diverse queries across multiple accounts and mixing them with less diverse queries. They do this to ensure that our coverage-aware defense does not flag the user as malicious due to the low embedding coverage. Further, the sybil adversary can gather these outputs from different accounts, effectively achieving a high embedding coverage, while still obtaining minimally perturbed outputs and evading detection. To mitigate the risk of sybil-based attacks, we introduce a defense that perturbs model outputs using a random affine transformation. While the random affine transformation remains consistent for a given legitimate user who queries the API from a single account, sybils obtain model outputs tampered by random affine transformations for every account they query the API with. In other words, we apply different affine transformations per account. To evaluate the effectiveness of

this defense, we simulate a downstream classification task by using the obtained model outputs. With the training of the classifier on consistently transformed model outputs, we observe that high utility is preserved for legitimate users, with minimal degradation. On the other hand, Sybil adversaries, who receive inconsistent model outputs transformed by random affine transformations from every account, observe a significant degradation in the utility, due to ineffective learning. Thus, we show that our defense also prevents sybil attacks by preserving the consistency of model outputs for legitimate users, while disrupting it for sybil adversaries. The results in the table below show the effectiveness of our defense against Sybil adversaries.

Table 5: **Downstream classification accuracy (%) for legitimate and sybil users under our affine transformation defense**. Legitimate users, with a single account, receive model outputs with a *shared* affine transformation across queries, preserving downstream utility. In contrast, sybil adversaries receive inconsistently transformed outputs across 4 accounts, severely degrading their model's performance. Results are shown for SST2, MovRev, and IMDB datasets.

| User Type | #Queries | Downstream Task | Downstream Accuracy |
|-----------|----------|-----------------|---------------------|
| LEGIT | 2000 | SST2 | 95.00 |
| SYBIL | 500 x 4 | SST2 | 50.60 |
| LEGIT | 2000 | MovRev | 88.80 |
| SYBIL | 500 x 4 | MovRev | 11.00 |
| LEGIT | 2000 | IMDB | 95.20 |
| SYBIL | 500 x 4 | IMDB | 4.80 |

## 8 CONCLUSIONS

We formalize the inversion from next-token probability vectors of prompt-tuned LLMs to soft prompts. We show that it is possible to recover functionally equivalent soft prompts using query-access to prompt-tuned LLMs, which achieve comparable model performance on different downstream tasks. Given the practical risks posed by this attack, we introduce Coverage Aware Perturbation (CAP), an active defense that tracks the embedding coverage of a potential attacker's diverse queries and penalizes too diverse queries while maintaining performance for benign users. Furthermore, we show that CAP is also successful in defending against recent last-layer extraction attacks in language models. Thereby, our defense successfully protects both ends of LLM APIs and contributes to their safe deployment.

**Reproducibility Statement.** We use all publicly available standard datasets and model architectures for experimental evaluation for our attack and CAP defense. Details regarding the same can be found in Section 4. We also provide details regarding hyperparameters in Appx. E. We also provide the code as a form of supplementary material during the submission.

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

## A  ADDITIONAL RELATED WORK

**Discrete Prompt Stealing.**  (Sha & Zhang, 2024) showed how to steal the text-based prompts. They motivate this attempt by arguing that organizations increasingly rely on carefully engineered prompts to elicit high-quality outputs from large language models (LLMs). This attack also aims to recover such proprietary discrete prompts using only the model's generated outputs. The proposed method comprises two key components: (1) inferring structural properties of the original prompt (*e.g.,* whether it is direct, role-based, or in-context), and (2) regenerating a prompt that closely resembles the original. Their results demonstrate that even subtle properties, such as the number of examples or specific instructions, can be reliably inferred and reconstructed. This work highlights a critical and emerging threat to the intellectual property and security of prompt engineering practices.

**LLM Data Extraction.** A recent line of work introduces a practical data extraction attack termed neural phishing  (Panda et al.). This attack enables adversaries to extract sensitive or personally identifiable information (PII), such as credit card numbers, from models trained on user data, achieving notably high success rates in some cases. Crucially, neural phishing operates under minimal assumptions: the attacker is only required to inject a small number of benign-looking sentences into the training corpus, guided merely by vague priors about the structure of the underlying user data. This highlights the risks posed even by subtle and plausibly deniable data poisoning strategies.

**Stateful Active Defenses against Model Stealing.** We also further elaborate on other type of active defenses that explicitly maintain a state of the users' queries, similarly as the passive defenses, but instead of stopping the service, they lower the quality of outputs. For example, the adaptive misinformation defense proposed by  (Kariyappa & Qureshi, 2020) aims to degrade model stealing attempts by identifying whether a query is ID or OOD. For OOD queries, the defense deliberately returns incorrect predictions.  While effective in reducing the accuracy of an attacker's stolen model, particularly when the attacker lacks access to ID samples, this approach also risks degrading performance for benign users. The defense relies on an OOD detector trained with both ID and OOD data. In a follow-up work,  (Kariyappa et al., 2021) propose an alternative defense that trains an ensemble of diverse models. This ensemble is designed to yield accurate predictions on ID queries while producing inconsistent or dissimilar outputs for OOD queries. A hash function, assumed to be secret, is used to select the appropriate model in the ensemble for each query. Both approaches, however, rely on prior knowledge of the attacker's OOD data, which is typically difficult to define in advance. As Hsu et al. (2020) note, the process of selecting representative OOD data can introduce significant bias, thereby limiting the robustness and generalizability of such defenses.

**Local Sensitive Hashing (LSH).** LSH is a probabilistic technique, introduced in  (Indyk & Motwani, 1998) and expanded in  (Datar et al., 2004), and hashes similar objects in similar hash buckets. In our work, we use random projection for cosine similarity. Each hash function $h_{\mathbf{r}}$ is defined as:

$$h_{\mathbf{r}}(\mathbf{x}) = \text{sign}(\mathbf{r}^{\top}\mathbf{x}),$$

This hash function maps the input vector $\mathbf{x} \in \mathbb{R}^d$ to one binary value depending on the sign of its projection along a randomly chosen direction $\mathbf{r} \in \mathbb{R}^d$. Using multiple random vectors, a hash code can be defined to place x in a specific bucket.

**Parameter-Efficient Fine-Tuning (PEFT).** PEFT methods are techniques that adapt large pre-trained language models (PLMs) to downstream tasks by fine-tuning only a small subset of parameters, thereby reducing computational cost, memory usage, and storage requirements while achieving superior performance. *Soft prompt tuning* is a widely used PEFT method in which a small number of trainable virtual tokens are prepended/added the input sequence. These tokens encode task-specific information. The virtual token embeddings are optimized during soft prompt tuning. The Hugging

Face `PEFT` library provides a framework for incorporating PEFT techniques within the Transformers ecosystem.

## B   ADDITIONAL EXPERIMENTS

Table 6: **Breakdown of Runtime (in seconds) of our Attack Pipeline using the Amazon task, with evaluation on YELP task.** The total attack cost includes (A) Distillation stage, with the following sub-steps: 1) Querying and obtaining responses from prompt-tuned LLM, with T5-base as the backbone architecture 2) Optimizing the randomly initialized soft prompt to mimic the prompt-tuned LLM's outputs, (B) Inversion stage, with the following substeps: 1) Training an inversion model on the model responses and extracted soft prompt embedding of AMAZON task 2) Using the trained inversion model to extract soft prompt for YELP task. The inversion stage includes reshaping the high-dimensional probability vector and further training it. We show that our inversion process offers an efficient alternative to tuning prompts from scratch, as the inversion cost is even lower than the cost to tune a prompt from scratch and further amortized when several soft prompts are inverted instead of fine-tuned.

| Attack Stage (AMAZON) | Runtime (seconds) |
|---|---|
| **A. Distillation** | |
| (1) Obtain softmax outputs for 1K text queries | 127.81 |
| (2) Optimization / extraction | 339.80 |
| **Total Distillation Runtime** | 467.61 |
| **B. Inversion** | |
| (1) Inversion model training | 20.60 |
| (2) Inversion | 0.0222 |
| **Total Inversion Runtime** | 20.6222 |
| **Complete Attack Runtime** | 488.2322 |
| **C. Prompt Tuning from Scratch** | |
| YELP | 11,021 |

Table 7: **Comparison of target and adversary LLMs' downstream performance when both of their underlying model architectures are different.** We report the downstream performance for the target prompt-tuned LLM and the adversary's prompt-tuned LLM, considering that both the tuned LLMs do not share the same underlying model architecture. It is observed that the downstream performance with adversary's tuned LLM is not only substantially higher than with randomly initialized prompt embedding, but is also closely matching the target prompt-tuned LLM.

| Task | Target LLM | Adversary LLM | Target (%) | Adversary(%) | Random (%) |
|---|---|---|---|---|---|
| AMAZON | T5-base | T5-small | 90.89 | 86.46 | 49.80 |
| YELP | T5-base | T5-small | 93.90 | 89.14 | 51.80 |
| SST-2 | T5-base | T5-small | 93.69 | 89.33 | 50.60 |
| MOVREV | T5-base | T5-small | 82.27 | 80.58 | 50.00 |
| AMAZON | T5-base | T5-large | 90.89 | 90.21 | 49.80 |

## C   TIMING-BASED SOFT PROMPT LENGTH EXTRACTION ATTACK

We conduct the first timing side-channel-based soft prompt length extraction attack, which precisely estimates the number of virtual tokens used by the prompt-tuned LLM in a black-box setup. We show that the seemingly unimportant metadata about the prompt-tuned model, like the soft prompt's length, can be leveraged to further expedite our extraction and inversion attacks against prompt-tuned LLMs.

Determining the exact length of the soft prompt employed by a prompt-tuned LLM is particularly challenging. On querying the API that hosts the prompt-tuned LLM in a black-box setup, users do not have access to the soft prompt embedding and length. Users can only access the model's

Table 8: **Extraction of functionally equivalent soft prompts with encoder-only model architecture.** We report the downstream performance for the target and adversary's prompt-tuned LLM, and target LLM with a randomly initialized soft prompt embedding. It is observed that for encoder-only architectures like roberta-base, the downstream performance with the extracted prompt is comparable to that with the target prompt.

| Dataset | Target Model | Target(%) | Adversary(%) | Random(%) |
|---------|--------------|-----------|--------------|-----------|
| SST-2 | roberta-base | 92.32 | 90.48 | 49.08 |
| IMDB | roberta-base | 91.07 | 90.12 | 49.88 |
| MOVREV | roberta-base | 85.37 | 81.24 | 50.00 |

Table 9: **Extraction of functionally equivalent soft prompts with partial probability access.** We report downstream accuracy (%) for the target prompt-tuned LLM, the adversary's prompt-tuned LLM, the target model with randomly initialized prompts, and the target model without soft prompts (zero-shot), assuming access to the top-$k$ probabilities. Both target and adversary models use T5-base as the backbone.

| Task | $k$ | Target (%) | Adversary (%) | Random (%) | Zero-shot (%) |
|------|-----|-----------|---------------|------------|---------------|
| | | | **Top-5 Probabilities** | | |
| YELP | 5 | 93.60 | 92.20 | 51.80 | 51.90 |
| AMAZON | 5 | 90.40 | 89.80 | 49.80 | 49.80 |
| MOVREV | 5 | 83.02 | 83.48 | 46.70 | 50.09 |
| | | | **Top-1 (Argmax Only)** | | |
| YELP | 1 | 93.60 | 51.50 | 51.80 | 51.90 |
| AMAZON | 1 | 90.40 | 49.70 | 49.80 | 49.80 |
| MOVREV | 1 | 83.02 | 46.80 | 46.70 | 50.09 |

outputs. However, even the most descriptive LLM output, like probability distribution over the vocabulary, does not inform the adversary about the precise length of the hidden soft prompt. This is because soft prompts are continuous embeddings that are prepended to the actual input text embeddings to form a concatenated input representation, which is collectively processed by the model's attention mechanism. This embedding sequence lacks an explicit demarcation between the soft prompt embeddings and input text embeddings, making it infeasible for an adversary who can solely observe outputs to estimate the length of the soft prompt. Moreover, (Lester et al., 2021) shows that the correlation between prompt length and performance is not linear beyond a certain length threshold, with performance gains plateauing after 20 virtual tokens. This makes the prompt length estimation more intractable, as similar downstream performances could result from vastly different prompt lengths. We circumvent these limitations by proposing the first timing-based side channel attack to determine the length of the soft prompt. The key insight behind this side-channel attack is that longer sequences require more time for the model to process, as also demonstrated in (Vaswani et al., 2017; Katharopoulos et al., 2020; Hiller et al., 2024; Zhang et al., 2023). By carefully analyzing how varying the length of the prompt influences the model's response latency, SPLIT effectively determines the length of the soft prompt in a black-box access setup.

**Problem.** We consider the problem of precisely extracting the length T of the soft prompt employed by the prompt-tuned LLM. Concretely, given a black-box query access to a prompt-tuned LLM whose underlying model architecture is known to an adversary, we aim to determine if an adversary can infer the length T of the prompt used by the LLM.

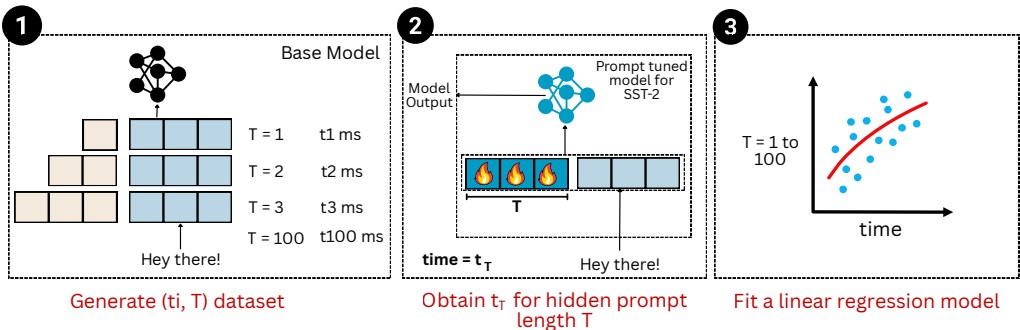

Figure 2: **Prompt length estimation for a prompt-tuned model in three stages** - i) Query a base model with random text prefixes by prepending soft prompt lengths ranging from 1 to 100 and build a dataset of response times ii) Query the prompt-tuned model with the same prefix and obtain response time iii) Fit a linear regression model on the data from step 1 to estimate the length of the prompt effectively.

## C.1 PROPOSED METHOD

As shown in Figure 2, our timing-based side channel attack works in three stages as detailed below:

**Generate a (response latency, prompt length) dataset.**   First, using the knowledge of the underlying architecture of the pre-trained LLM, an adversary intends to obtain the API response time when prompt embeddings with varying lengths are prepended to the constant arbitrary text prompt embeddings. As API response time is independent of whether or not the soft prompt is randomly initialized or tuned on a specific downstream task, only randomly initialized embeddings are utilized for this attack. Although the response time primarily comprises the time to process the input embeddings (soft prompt and text), environmental factors such as memory scheduling and network transmission delays influence the response time. These external factors also introduce noise that distorts the actual response time  (Zheng et al., 2024a). To overcome this limitation, the same arbitrary text prefix is queried to the model $n$ times for each prompt length, and the response times are aggregated to nullify the effect of these environmental factors. Importantly, we do not query the prompt-tuned LLM that we have black box access to. Instead, we access a local copy of the pre-trained LLM and prepend soft prompt embeddings of varying lengths to the input text embeddings to compute response time per prompt length. This allows us to obtain response latency as a function of the soft prompt length. Although there is no restriction on the maximum prompt length to be prepended to generate a dataset, we constrain it to 100 tokens, particularly because it has been shown in  (Lester et al., 2021) that beyond 100 tokens, the downstream performance degrades for large models, which limits the use of $> 100$ virtual tokens in realistic scenarios. Besides that, soft prompts released publicly  (Research, 2021) also do not exceed 100 virtual tokens, further strengthening our assumption.

**Obtain response latency for hidden prompt.**   Assuming a black-box query access to the prompt-tuned LLM wherein the soft prompt embedding and its length is hidden to the adversary, we query the same arbitrary text prefix $n$ times, similar to stage 1, to the prompt-tuned LLM and note the API response time.

**Fit a linear regression model to extract T.**   We fit a simple linear regression model, based on the monotonic relationship between prompt length and API response latency obtained in stage 1, and predict the length of soft prompt T.

Using these three stages, this attack effectively determines the length of the soft prompt as shown in Figure 3.

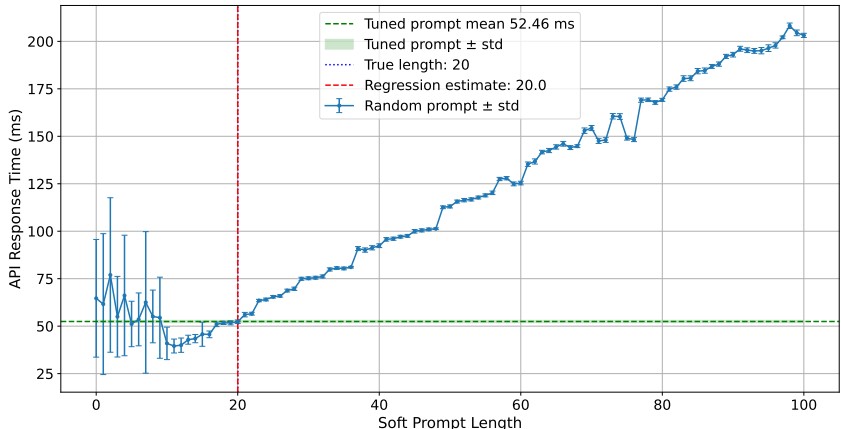

Figure 3: **Estimating the length of the soft prompt via timing-based side channel attack.** We present API response latency as a function of the number of virtual tokens/soft prompt length used. We observe that there is a monotonic relationship between the soft prompt length and API response time, and that it is possible to determine the exact length of the soft prompt, hidden from an adversary.

## D  STATE-OF-THE-ART PRODUCTION LANGUAGE MODEL EXTRACTION ATTACKS

Our proposed defense CAP defends against the state-of-the-art extraction attack by (Carlini et al., 2024). In this section, we discuss more about this extraction attack. This work by (Carlini et al., 2024) highlights the security risk of such an attack on production language models like OpenAI's GPT, Google's PaLM, etc. The attack primarily focuses on extracting information such as dimensionality and final layer weight matrix from production language models. The work emphasizes that despite these models being accessible to users through black-box APIs, significant information about LLMs—like hidden dimensionality and the last-layer weight matrix—can be successfully extracted. Their attack mainly exploits the API features that reveal the output logits of the model, thereby recovering critical information about the model architecture.

**Attack Intuition and Methodology.** We first discuss the attacker's threat model. The attacker has black box access to a production language model, meaning the attacker can query the LLM via the API but does not have access to information about the model architecture, including weights, training data, etc. The attacker sends a large number of queries to the LLM and analyzes the obtained model outputs, in the form of logits. The key insight about the attack is that the last layer of the transformer model, which maps the hidden states to logits, is a linear transformation that typically has a low rank. This final layer can be approximated using the output logits collected by the attacker. Due to the low-rank structure of the final layer, diverse or linearly independent queries sent by the attacker explore new directions in the embedding space. With enough linearly independent or diverse queries, the attacker can recover the dimensionality of several production language models and further extract the final-layer weight matrix of the transformer model. While the work explores different levels of API access, including full logits and top-k logits, we primarily focus on the scenario where the API exposes full logits.

Empirical evidence demonstrates that this attack approach is quite effective and results in very low reconstruction error between the original and recovered last layer weight matrix. Additionally, this approach reveals an almost accurate dimensionality of several production language models.

## E  DETAILS ON HYPERPARAMETERS

We present the details on hyperparameters for our attack and CAP defense in this section. We perform our experiments with Python 3.13, Pytorch, and use a server with 4X NVIDIA A100 GPUs.

Table 10: **Breakdown of Runtime (in seconds) of our defense pipeline using the AMAZON task.** We quantify the computational overhead for each of the components of the CAP defense pipeline to assess the efficiency and scalability of our defense. As a prompt-tuned LLM provider, this computational expense is significantly less than tuning a prompt from scratch in the bigger workflow.

| Defense Stage | Runtime (seconds) |
|---|---|
| (1) Coverage computation | 171.47 |
| (2) Spread computation | 0.87 |
| (3) Noise perturbation | 0.66 |
| **Total Defense Runtime** | 378.03 |

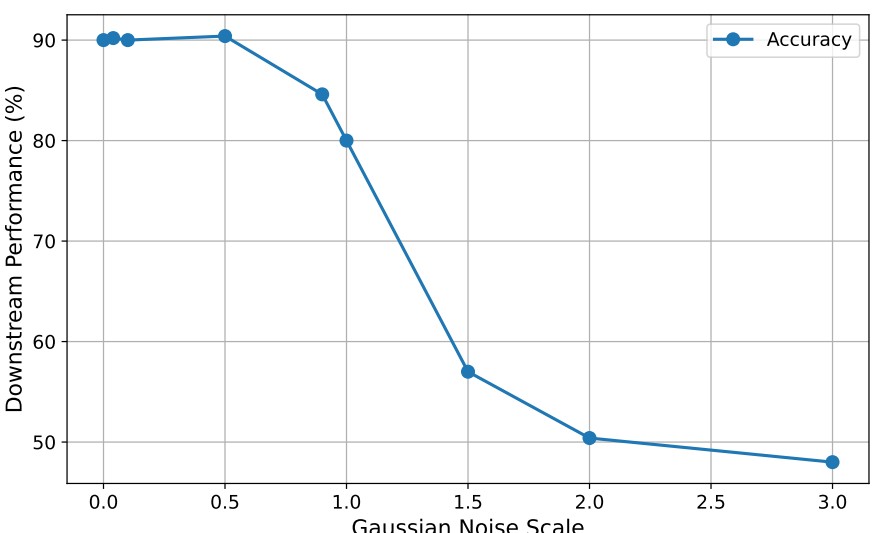

Figure 4: **Downstream performance decreases as the soft prompt is perturbed more.**

In our experiments on in-distribution queries, the adversary assumes to have knowledge of the distribution from which the prompt tuning data comes. Thus, if we consider a downstream task, Amazon Polarity, the adversary only queries the prompt-tuned LLM with this dataset. However, in our experiments on OOD queries, the adversary's prompt-tuned LLM is distilled from the teacher using text queries sampled from different downstream datasets (Amazon Polarity, YELP, IMDB, AG News, Rotten Tomatoes movie reviews, SST-2, DBpedia, SNLI, MNLI, TweetEval, and synthetic text queries). We use a batch size of 32 and a learning rate of $5 \times 10^{-3}$ and 30 epochs. The maximum sequence length is set to 128 tokens. During inversion, we apply a linear transformation function to the prompt extracted from the distillation stage, as shown in Figure 1. This is to ensure that the model does not overfit to the extracted prompt embedding. The Gaussian noise added has a noise scale of 0.1, ensuring that the downstream performance remains unaffected. This observation also comes from Figure 4. For our CAP defense against soft prompt extraction attacks, we set the number of buckets to $2^{11}$, which equals 2048 buckets. The baseline noise $\lambda$ is set to 0.0005. The scaling factor $\alpha$ is chosen as 8.0, while the parameter $\beta$ is set to 0.19. The batch size for computing coverage and spread is 100. Additionally, the weights for the different components of the defense are specified as follows: the coverage weight is 0.05, the novelty weight is 0.35, and the spread weight is 0.45. The LLM provider sets these values as per the desired degree of penalization.

**Choice of T5 Backbone.** Our entire pipeline is based on the T5 architecture. T5 uses a pure text-to-text framework: every task—classification, sentiment analysis, NLI passes through the same decoder interface, thereby providing the next-token probability distribution. This uniformity allows us to design a single inversion model that could used across multiple tasks. Choosing T5 therefore provides architectural simplicity consistent with widely deployed prompt-tuned systems.

Table 11: **Summarization results on CNN and ArXiv.** We report ROUGE-1/2/L scores of the target prompt-tuned LLM (teacher) and evaluate reconstructed and random prompts on the same datasets.

| Task | Target LLM (Teacher) | Reconstructed Prompt | Random Prompt |
|------|---------------------|---------------------|---------------|
| *ROUGE-1 / ROUGE-2 / ROUGE-L* | | | |
| CNN | 0.2621 / 0.1035 / 0.1928 | 0.1191 / 0.0460 / 0.0874 | 0.1530 / 0.0591 / 0.1142 |
| ArXiv | 0.1726 / 0.0461 / 0.1156 | 0.0212 / 0.0061 / 0.0148 | 0.1713 / 0.0449 / 0.1150 |

**Hyperparameters for CAP defense against last-layer extraction attack.** To query the prompt-tuned LLM, we use a set of prompts curated in Paleka (2025). Additionally, we use a gap threshold of 5.0 and a minimum singular value of 1e-6.

## F RUNTIME FOR OUR INVERSION ATTACK AND CAP DEFENSE

We present insights into the overhead that the attack and the CAP defense may introduce in Table 6 and Table 10, respectively. First, we compute the prompt tuning time with T5-base based on a representative task, YELP, and find that it takes approximately 11,021 seconds in total. However, our attack pipeline, based on AMAZON, takes significantly less time (488.23 seconds) than tuning a prompt from scratch to extract a functionally equivalent soft prompt of unseen downstream task YELP.

**The Use of Large Language Models.** In this work, we acknowledge that Large Language Models (LLMs) were used for exploring research literature related to the topic of the paper and to polish the writing. Additionally, we utilized LLMs as one of the sources for verifying and debugging our experimental implementation.

