# OpenReview forum: "Stealing and Defending the Ends of LLMs"
_ICLR.cc/2026/Conference — Submitted to ICLR 2026_

### Official Review · Reviewer_poZL · 2025-10-25

**Soundness:** 2
**Presentation:** 2
**Contribution:** 2
**Rating:** 2
**Confidence:** 4

**Summary:**

The paper studies the security of soft-prompt–tuned LLMs exposed via APIs that return next-token probabilities. It presents a two-stage attack: (i) distillation, which learns a functionally equivalent soft prompt by minimizing KL divergence between victim and surrogate outputs; and (ii) inversion, which trains a Transformer to map probability vectors directly to soft-prompt embeddings and claims cross-task generalization. To mitigate, the paper proposes CAP, an active defense that estimates query coverage in embedding space (via LSH bucket coverage, new-bucket rate, and spread) and injects Gaussian noise into the served prompt or logits in proportion to coverage. Experiments on T5 variants (plus some roberta-base) show strong Stage-1 performance; Stage-2 reports transfer across related tasks; CAP reduces extraction and increases last-layer RMSE. Additionally, a timing side-channel (SPLIT) for estimating prompt length appears in the appendix.

**Strengths:**

- S1: KL-based distillation on black-box probabilities seems simple, effective, and broadly applicable.
- S2: The authors target a realistic API settings (full/top-k probabilities), and frame risks/defenses at both the input (soft prompt) and output (last layer) ends.
- S3: Results show some transferability on T5 sizes and roberta-base; defense also touches Pythia/GPT-2 for last-layer extraction.
- S4: The separation into distillation and inversion clarifies assumptions and makes the attack path easy to reason about.

**Weaknesses:**

- W1 (Baselines): The Random prompt in Table 1 hovers around $50$% (even for MNLI where random chance should be $33$%?), which is hard to interpret. It would be great to include as a baseline T5-base with no soft prompt or a discrete prompt. This would clarify whether random soft prompts actively harm the model.
- W2 (Missing comparative baselines): Beyond ``random'', I believe this paper needs at least one additional baseline to contextualize the performance. For example this could be [Morris et al,. 2024: Language Model Inversion] to recover a discrete text prompt and compare its downstream performance to the presented results.
- W3 (Defense utility & adaptivity): I am skeptical about the assumption that benign queries are low-diversity (e.g. multi-task users). As this seems to be a fundamental assumption for the proposed defense method, it would be great to justify this with actual user data (e.g. WildChat dataset).
- W4 (CAP results interpretation): Table 2 shows strong perturbation on the attack queries; reporting per-run coverage metrics (C/N/S) would explain why these are deemed diverse. Additionally, in table 3, it would be important to add a row for LEGIT + CAP-OFF to show absolute utility drop.
- W5 (Model choice): T5 seems a bit dated by now. Running some of the T5 experiments in stage 1 with e.g. Pythia would improve this.

In its current state I cannot recommend acceptance of the paper. However, the points I raised are adequately addressed, I am willing to increasing my score.

**Questions:**

- Q1: Could you clarify how exactly the LEGIT/ATTACK classification in Table 3 is defined?
- Q2: The adversary sometimes even outperforms the target (Table 2), and in summarization (Table 10) random $\approx$ target while reconstructed underperforms. Do you have an explanation for these?It might be good to add multi-seed means and standard deviations to make the findings more trustworthy.
- Q3: If trained on a single task, it is unclear to me why the inversion model should generalize? A small ablation varying the number/diversity of training tasks, and a naive cross-task baseline (i.e. reuse the $D_1$ recovered sot prompt on $D_2$) would clarify the incremental value.

---

> ### Author Response · Authors · 2025-11-21
> **Rebuttal**
>
> We thank the Reviewer for the positive and detailed feedback. We appreciate that the Reviewer finds our attack broadly applicable, realistic and recognizes that our defense is applicable across models like GPT-2. We provide detailed answers to questions below
>
> >**W1 (Baselines): It would be great to include as a baseline T5-base with no soft prompt or a discrete prompt. This would clarify whether random soft prompts actively harm the model.**
>
> As per your suggestions, we have included this baseline in tables 1, 2, etc in the updated manuscript. The results suggest that the zero-shot performance of the model is quite comparable to that with a random soft prompt.
>
> >**W2 (Missing comparative baselines): Beyond ``random'', I believe this paper needs at least one additional baseline to contextualize the performance. For example this could be [Morris et al,. 2024: Language Model Inversion] to recover a discrete text prompt and compare its downstream performance to the presented results.**
>
> While Morris et al. 2024 demonstrates impressive discrete prompt recovery, our attack targets soft prompt embeddings in the continuous embedding space. To address the baseline concern, we consider two baselines - zero shot performance (model performance without soft prompt) and model performance with randomly initialized soft prompts, to demonstrate the effectiveness of our attack, when recovered soft prompt is used for determining the performance of the model.
>
> >**W3 (Defense utility & adaptivity): I am skeptical about the assumption that benign queries are low-diversity**
>
> As we intend to defend a prompt-tuned model accessible via an API endpoint, against inversion attacks, our assumption is primarily architectural. Task-specific APIs provide prompt tuned models for specific tasks, thus benign users access specific API only for the intended task. Multi-task users can use separate API endpoints to obtain high performance for different tasks. However, to extract a soft prompt that generalizes across tasks, attackers must systematically explore the embedding space with queries that are more diverse than task specific queries. Our defense extends the successful B4B paper [Ref 1], which distinguishes legitimate image encoder users from adversaries using embedding-space coverage.  [Ref 1] Dubiński, Jan, et al. “Bucks for Buckets (B4B): Active Defenses Against Stealing Encoders” NeurIPS 2023
>
> >**Could you clarify how exactly the LEGIT/ATTACK classification in Table 3 is defined?**
>
> The Coverage-Aware Perturbation (CAP) defense mechanism distinguishes between legitimate users (LEGIT) and adversaries (ATTACK) primarily by monitoring the diversity of their queries and the resulting coverage of the LLM's input embedding space. Legitimate Users (LEGIT) typically queries a prompt-tuned LLM to solve a specific downstream task. Their queries remain task-focused, meaning they gradually explore the embedding space but do not aggressively cover a major part of it. Adversaries (ATTACK) intend to steal the soft prompt's functionality, query the LLM with highly diverse inputs.
>
> >**The adversary sometimes even outperforms the target (Table 2), and in summarization (Table 10) random  target while reconstructed underperforms. Do you have an explanation for these?**
>
> Due to the existence of multiple near-optimal soft prompts that lie in the same low-dimensional subspace, the extracted prompt  may, in some scenarios, result in better performance than the original prompt. To clarify if this, we experiment with 4 different seeds (apart from the one in the paper). After doing variance analysis for MOVREV in Table 2, we find that the Adversary (CAP Off) performance is 81.81% which is less than and comparable to the target 82.10%. The high MOVREV accuracy for adversary for one seed is a consequence of the prompt-augmentation during inversion. This behavior where the adversary exceeds the target model is not observed across all seeds (45, 100, 1335, 50) with target accuracy 82.10 and adversary’s accuracy (80.40, 79.10, 79.70 and 80.52) with standard deviation of 4.24.
>
> >**If trained on a single task, it is unclear to me why the inversion model should generalize?**
>
> The key insight for the generalization of the inversion model is that AMAZON is inherently more diverse than smaller tasks like SST-2 , making it ideal task for inversion training. In Stage 1, the AMAZON dataset allows the soft prompt extraction to capture diverse probability vectors that encode rich signal. When training the inversion model on these AMAZON probability vectors and extracted soft prompt embeddings, the model learns universal architectural patterns rather than task-specific patterns. Since smaller downstream tasks occupy narrower input distributions that are subsets of AMAZON's diversity, the inversion model generalizes. Table 2 validates this, demonstrating that task diversity is critical for generalization.

---

> > ### Comment · Reviewer_poZL · 2025-11-24
> > **Follow-up question**
> >
> > Thank you for your response and clarifications, I will take them into account in my final rating.
> >
> > I have one additional follow-up question on the inversion model, as I want to make sure I understand this fully. How exactly does the dataset look like for training the inversion model? To me it seems that you would distill a single soft prompt on e.g. the AMAZON dataset, which gives a dataset consisting of N probability vectors and one target soft prompt. But I assume a dataset for training the inversion model should have multiple targets, so where do we get these from? Do you repeat the distillation for multiple seeds, do you treat multiple task datasets in parallel, or do you partition the dataset into multiple subsets and extract a soft prompt for each subset? This was not clear to me from the manuscript, and I would highly appreciate a clarification on this.

---

> > > ### Author Response · Authors · 2025-11-25
> > >
> > > We thank the reviewer for the insightful question. We are happy to clarify how we obtain multiple targets for training the inversion model.
> > >
> > > First, we do not (i) repeat distillation with multiple seeds, (ii) partition a single dataset, or (iii) run distillation in parallel. Instead, we perform one distillation per downstream task, so each task (e.g., AMAZON, YELP, etc) yields one extracted soft prompt. The task-specific extracted prompt serves as a target needed for training the inversion model.
> > >
> > > But as correctly noted by the reviewer, our distillation stage (Stage 1) results in N probability vectors mapped to a single target soft prompt. To address this data scarcity problem, we apply data augmentation to the recovered target prompt by applying a linear transformation function combined with Gaussian noise perturbation. By creating augmented versions of the extracted prompt from Stage 1, we expand the training dataset from a single (prob, prompt) pair into many pairs. This produces additional prompt variants while preserving downstream functionality, ensuring that the inversion model learns robust features rather than overfitting to a single embedding, as validated in Table 2. We mention about this in Appendix E and also mention f as the linear transformation function in Figure 1, during the inversion stage. We include the lines from Appendix E for your reference below.
> > >
> > > During inversion, we apply a linear transformation function to the prompt extracted from the distillation stage, as shown in Figure 1. This is to ensure that the model does not overfit to the extracted prompt embedding. The Gaussian noise added has a noise scale of 0.1, ensuring that the downstream performance remains unaffected.
> > >
> > >
> > > We hope this explanation clarifies the reviewer's concern.

---

### Official Review · Reviewer_n4Zu · 2025-10-27

**Soundness:** 3
**Presentation:** 2
**Contribution:** 1
**Rating:** 2
**Confidence:** 4

**Summary:**

This paper introduces a technique to distill and invert prompt tuned models through soft prompts.
From the system provider side, the paper also introduces a defense against the proposed inversion attack through monitoring the adversaries' query diversity.
Experiment are carried out on small scale models and datasets showing promising results.

**Strengths:**

The main strengths of this work are:

1) The proposed approach is generic. Distilling the inverting through soft prompts does not assume any specific architecture nor dataset.

2) The related work section in this paper is thorough.

3) The proposed attack is easy to implement and provides good results under small scale experiments.

**Weaknesses:**

Despite the aforementioned strengths, this paper has major weaknesses that need to be addressed before getting it published.

1) Experiments are conducted under small scale models and datasets. It is hard to measure actual usefulness. To demonstrate the effectiveness of the proposed attack, experiments should be carried out on realistic size models (such as Llama3-7B), and on realistic tasks/datasets.

2) The assumption that the architecture is shared seems to be restrictive. It is often the case that the API does not provide any information about the architecture. Experiments should also include the case where there is a mismatch between the architecture of the target and the victim models.

3) The baseline (LLM with random SP initialization) is not convincing. At least, the performance of the base LLM without any SP should be included as a weak baseline. Further, the impact of the learnt soft prompts on different tasks (e.g. reasoning/coding/solving math problems) should be measured.

4) Experiments with comparison with other PEFT methods are missing (e.g. LoRA). While I understand the advantages of soft-prompts compared to LoRA, the argument in this paper would have been significantly strengthen if a comprehensive comparison against other PEFT methods is included.

5) The writing of this paper can be vastly improved. Many parts (such as Figure 1, Section 5.2) are very hard to parse and overcomplicates the simple message of this paper. For example, Equation (2) can be simplified to $TotalCost= \lambda + w_c (\frac{\alpha C}{\lambda \beta} -1 ) + \alpha w_n N + \alpha w_s \min (\frac{S}{S_{max}}, 1)$. Further, the tables can be located in the page they were mentioned in to ease the reading of the manuscript.

6) The proposed defense CAP is both naive and makes unrealistic assumptions. For instance, such defense can be surpassed with multiple attackers setting (each attach does not diversify the queried topics, but different attacks query different buckets).

7) The paper mentions that the inversion accelerates the distillation (e.g. in line 361), without providing a comparison nor quantification of the time/cost saved. A discussion along these lines is necessary with its corresponding experiments.

Overall, while I extremely appreciate the practicality of the proposed method, I believe that this work has to address significant concerns, provide important extra experiments and ablations before getting published.

**Questions:**

Please refer to the weaknesses section

---

> ### Author Response · Authors · 2025-11-21
> **Rebuttal**
>
> We thank the Reviewer for the positive and detailed feedback. We appreciate that the Reviewer finds our work generic, thorough and supported by good experimental results. We provide detailed answers to the concerns and questions in-line below
> Weaknesses:
>
> >**It is often the case that the API does not provide any information about the architecture. Experiments should also include the case where there is a mismatch between the architecture of the target and the victim models.**
>
> Table 7 in the paper (In Appendix) presents a comparison of target and adversary LLMs’ downstream performance when both of their underlying model architectures are different. We show through the experiments that our inversion attack is successful even when the architectures are different. We include that below for your reference.
>
> | Task | Target LLM | Adversary LLM | Target (%) | Adversary (%) | Random (%) |
> |--|--|---|----|---|---|
> | AMAZON | T5-base | T5-small  | 90.89 | 86.46 | 49.80 |
> | YELP   | T5-base | T5-small | 93.90 | 89.14 | 51.80 |
> | SST-2  | T5-base | T5-small | 93.69 | 89.33 | 50.60 |
> | MOVREV | T5-base | T5-small | 82.27 | 80.58 | 50.00 |
> | AMAZON | T5-base | T5-large| 90.89 | 90.21 | 49.80 |
>
> >**At least, the performance of the base LLM without any SP should be included as a weak baseline.**
>
> We agree that adding a baseline for a zero-shot setting would provide more insights on the effectiveness of our attack. We have accordingly included this baseline in the tables 1, 2 and others in the updated manuscript.
>
> >**Experiments with comparison with other PEFT methods are missing (e.g. LoRA). While I understand the advantages of soft-prompts compared to LoRA, the argument in this paper would have been significantly strengthen if a comprehensive comparison against other PEFT methods is included.**
>
> While we acknowledge that our work could be extended for other PEFT methods, our attack specifically targets recovering functionally equivalent soft prompt and its properties that they are prepended continuous embeddings that remain fixed at inference, making them extractable via distillation and inversion of model outputs such as probability vectors. LoRA and other PEFT methods inject trainable parameters throughout the model layers, not in the input embedding space. Thus, our inversion model is trained to map probability vectors to soft prompt embeddings, but this mapping does not transfer to LoRA weights or other distributed PEFT parameters embedded in hidden layers. Therefore, we omit the experiments on other PEFT methods here.
>
> >**The writing of this paper can be vastly improved. Many parts (such as Figure 1, Section 5.2) are very hard to parse and overcomplicates the simple message of this paper. For example, Equation (2) can be simplified. Further, the tables can be located in the page they were mentioned in to ease the reading of the manuscript.**
>
> We simplified the equation (2) as per your valuable suggestion. In the updated manuscript, we have best tried to locate the tables on the page they are mentioned/described for ease of reading.
>
> >**The proposed defense CAP is both naive and makes unrealistic assumptions. For instance, such defense can be surpassed with multiple attackers setting (each attach does not diversify the queried topics, but different attacks query different buckets).**
>
> We acknowledge that sybil adversaries can indeed fool CAP and hence also provide a defense against them. We also added the results from section 7 on Sybils below for your reference. This was also added to the main part of the paper in the updated version of our submission as Section 7.
>
> | User Type | # Queries  | Downstream Task | Downstream Accuracy (%) |
> |---|----|---|---|
> | LEGIT     | 2000          | SST2    | 95.00    |
> | SYBIL     | 500 × 4       | SST2    | 50.60   |
> | LEGIT     | 2000          | MovRev   | 88.80  |
> | SYBIL     | 500 × 4       | MovRev  | 11.00  |
> | LEGIT     | 2000          | IMDB | 95.20  |
> | SYBIL     | 500 × 4       | IMDB  | 4.80   |
>
> >**The paper mentions that the inversion accelerates the distillation (e.g. in line 361), without providing a comparison nor quantification of the time/cost saved. A discussion along these lines is necessary with its corresponding experiments.**
>
> In Appendix B, Table 6 and Table 10, we present a detailed overview and breakdown of the runtime of our attack and defense pipeline to show that the inversion accelerates distillation. We show that our inversion process offers an efficient alternative to tuning prompts from scratch, as the inversion cost is even lower than the cost to tune a prompt from scratch and further amortized when several soft prompts are inverted instead of fine-tuned. We also quantify the computational overhead for each of the components of the CAP defense pipeline to assess the efficiency and scalability of our defense. As a prompt-tuned LLM provider, this computational expense is significantly less than tuning a prompt from scratch in the bigger workflow.

---

### Official Review · Reviewer_VEkW · 2025-10-27

**Soundness:** 3
**Presentation:** 2
**Contribution:** 2
**Rating:** 2
**Confidence:** 4

**Summary:**

The paper presents a systematic framework for auditing LLMs' susceptibility to generating malicious code, demonstrating significant real-world impact through the discovery of active scam sites. However, the analysis of model-specific vulnerabilities lacks depth, and the scope is limited to URL-based threats, potentially overlooking other malicious code vectors. The guardrail evaluation is limited to a single system without exploring alternative defenses (Sec. 6.1, Sec. 6.2).

**Strengths:**

* Novel Attack Formulation
  - Introduces a two-stage attack combining distillation (KL divergence minimization, Eq. 1) and inversion, enabling cross-task prompt extraction (Sec. 3).
  - Demonstrates generalization to unseen tasks (e.g., 87.2% vs. 88.8% on YELP when trained on AMAZON, Table 2), reducing computational costs vs. tuning from scratch (Table 5).
  - Validated on diverse architectures (T5/Roberta) and tasks (classification/NLI), showing robustness to OOD queries (Table 1).

* Practical Defense Design
  - CAP leverages embedding-space coverage metrics (bucket coverage, spread) to distinguish adversarial vs. benign users (Sec. 5.1).
  - Perturbs prompts/outputs based on coverage (Eq. 2), reducing stolen utility to random levels (Table 3) while preserving benign user performance (e.g., 86.7% vs. 91.08% on AMAZON).
  - Extends to defend against last-layer extraction (RMSE increase from 1.96e-5 to 18.21 for T5-base, Table 4).

* Comprehensive Experimental Validation
  - Tests 4 LLMs in 2024 with varied Prompt/Codegen LLM combinations (Table 1), demonstrating robustness across model pairs and architectures.
  - Applies benchmark to 7 state-of-the-art 2025 models (Table 2), showing consistent vulnerability across diverse providers and model sizes.

**Weaknesses:**

* Limited Threat Model Realism
  - Assumes full access to next-token probabilities, but real-world APIs (e.g., OpenAI) often expose only top-k tokens (Sec. 4.1). Experiments with top-5 access (Table 8) show minor degradation, but broader constraints (e.g., rate limits) are unexplored.
  - Ignores ethical implications: No discussion of misuse risks (e.g., IP theft) or mitigation beyond CAP (Sec. 7).

* Inadequate Analysis of Guardrail Effectiveness
  - Only tests one guardrail (NeMo Guardrails), with no comparison to alternative safety mechanisms or discussion of why it failed.
  - No investigation into specific policy gaps (e.g., S24: "Use of scam API/website" in Fig. 9) that might improve detection.
  - Fails to address why guardrails missed all malicious outputs despite clear scam API references, leaving mitigation strategies unexplored.

* Insufficient Discussion of Training Data Poisoning Mechanisms
  - Claims data poisoning but provides no evidence of how poisoning occurred (e.g., specific training data sources or crawl processes).
  - Hypothesizes OpenAI models’ higher malicious rates due to "more extensive data containing scam-related content" (Sec. 6.1) without supporting evidence or analysis.
  - No investigation into why certain models (e.g., gpt-4o-mini) consistently produce higher malicious rates across combinations (Table 1), limiting understanding of root causes.

**Questions:**

1. In Section 1, the authors state that "the actual rate of malicious code generation likely exceeds this figure when considering attack vectors beyond URLs." Could the authors provide a preliminary analysis of other malicious code vectors (e.g., backdoors or worms) or discuss how their framework could be extended to include them? (Sec. 1)

2. In Table 1, the authors report varying malicious rates based on Prompt and Codegen LLM combinations. Could they provide more details on how specific Prompt LLM characteristics (e.g., keyword diversity or prompt specificity) correlate with higher malicious rates? (Sec. 6.1)

3. The paper claims "malicious content contamination is an industry-wide problem persisting despite advances in safety alignment" (Sec. 8). Could the authors discuss potential training data sanitization strategies that could mitigate this issue, and whether any such strategies were tested in their framework? (Sec. 8)

---

> ### Author Response · Authors · 2025-11-16
> **This review is not relevant to the content and scope of our paper**
>
> Dear AC and PC Members,
>
> We believe that this review is not relevant to the content and scope of our submission. Could you please advise us on how to proceed in this situation?
>
> Thank you for your help,
>
> Authors of submission #1529

---

> ### Author Response · Authors · 2025-11-18
> **Thank You for Updating Your Review**
>
> Dear Reviewer VEkW,
>
> Thank you for updating your review to reflect the content and scope of our submission. We appreciate it.
>
> With kind regards,
>
> The authors of submission #1529

---

> ### Author Response · Authors · 2025-11-21
> **Rebuttal**
>
> We thank the Reviewer for the positive and detailed feedback. We appreciate that the Reviewer finds our two-staged distillation and inversion attack well motivated, experimentally supported, recognizes our defense to be realistic, and overall finds our work on soft prompting security important. We provide detailed answers to the concerns and questions in-line below
>
> >**The evaluation does not deeply explore adversaries who modulate query diversity to evade CAP, limited analysis of adaptive attackers. How would CAP respond to such adaptive strategies?**
>
> We acknowledge that adaptive sybil adversaries can indeed fool CAP and hence also provide a defense against them. We also added the section on Sybils for your reference: We consider an adversary who queries the API from $n$ accounts. For every account, the model outputs (probabilities), which are released to the adversary. The adversary collects these model outputs to train the inversion model. Sybil adversaries try to circumvent our defense by carefully partitioning their diverse queries across multiple accounts and mixing them with less diverse queries. They do this to ensure that our coverage-aware defense does not flag the user as malicious due to the low embedding coverage. Further, the sybil adversary can gather these outputs from different accounts, effectively achieving a high embedding coverage, while still obtaining minimally perturbed outputs and evading detection.
> To mitigate the risk of sybil-based attacks, we introduce a defense that perturbs model outputs using a random affine transformation. While the random affine transformation remains consistent for a given legitimate user who queries the API from a single account, sybils obtain model outputs tampered by random affine transformations for every account they query the API with. In other words, we apply different affine transformations per account. To evaluate the effectiveness of this defense, we simulate a downstream classification task by using the obtained model outputs. With the training of the classifier on consistently transformed model outputs, we observe that high utility is preserved for legitimate users, with minimal degradation. On the other hand, Sybil adversaries, who receive inconsistent model outputs transformed by random affine transformations from every account, observe a significant degradation in the utility, due to ineffective learning. Thus, we show that our defense also prevents sybil attacks by preserving the consistency of model outputs for legitimate users, while disrupting it for sybil adversaries. The results in the table below show the effectiveness of our defense against Sybil adversaries.
>
> | User Type | # Queries     | Downstream Task | Downstream Accuracy (%) |
> |-----------|------|---------|------|
> | LEGIT     | 2000  | SST2       | 95.00   |
> | SYBIL     | 500 × 4   | SST2      | 50.60  |
> | LEGIT     | 2000 | MovRev  | 88.80 |
> | SYBIL     | 500 × 4  | MovRev  | 11.00|
> | LEGIT     | 2000  | IMDB  | 95.20   |
> | SYBIL     | 500 × 4 | IMDB  | 4.80  |
>
> >**The attack is shown to work with top-k probabilities, but how does performance degrade as k decreases, especially down to top-1 output, which many APIs enforce?**
>
> When restricted to top-1 probabilities, our inversion model yields random accuracy. This is indeed one of the limitations of our attack and this result is expected as providing only the argmax class collapses almost all information about the underlying probability vector. Since inversion relies on the probability distribution, the mapping becomes non-invertible. This mirrors observations in prior work on black-box model extraction: label-only access is insufficient to recover continuous parameters or functionally equivalent adaptations. Several major LLM API providers expose top-k token probabilities: OpenAI's Chat Completions API supports top_logprobs (0-5), Together AI supports top_logprobs (0-20), and Google's Gemini on Vertex AI supports logprobs (1-20), making our attack relevant in practice. We have updated the Table 9 in the manuscript to include the performance for top-1 outputs, and we include it below for your reference.
>
> | Task   | k | Target (%) | Adversary (%) | Random (%) |
> |--|---|----|--|---|
> | YELP   | 1 | 93.60 | 51.50 | 51.80 |
> | AMAZON | 1 | 90.40 | 49.70 | 49.80 |
> | MOVREV | 1 | 83.02 | 46.80 | 46.70 |
>
> >**The description of the inversion model is brief; more clarity on architecture choices and failure modes would help interpret its capabilities.**
>
> In Appendix E of the updated manuscript, we present more details on the inversion model.
>
> >**Could an adaptive attacker interleave benign-looking task-specific queries with diverse probing queries to lower their coverage signal?**
>
> Adaptive adversaries can indeed try to fool the CAP defense by interleave task-specific queries with diverse queries, however, CAP defends against Sybil attacks using random affine transformations, as presented in the previous response.

---

> > ### Comment · Reviewer_VEkW · 2025-11-24
> > **Reviewer's response**
> >
> > I appreciate the the authors’ detailed rebuttal.  These responses address several of my concerns and strengthen the paper’s practical relevance. I will take your rebuttal into account when forming my final recommendation.

---

> > > ### Author Response · Authors · 2025-11-25
> > >
> > > We thank the reviewer for the thoughtful consideration of our rebuttal and for taking our clarifications into account in the final evaluation.

---

### Official Review · Reviewer_1dpA · 2025-10-31

**Soundness:** 2
**Presentation:** 3
**Contribution:** 3
**Rating:** 4
**Confidence:** 3

**Summary:**

The paper studies the vulnerability of soft-prompt-tuned LLM deployments and proposes: (1) Attack: A two-stage black-box attack, including distillation stage (learn a functionally equivalent soft prompt by minimizing KL divergence to the victim’s next-token probabilities) and inversion stage (train a small transformer to map next-token probability vectors to soft-prompt embeddings that transfer across tasks); (2) Defense: Coverage-Aware Perturbation (CAP), track query diversity via LSH-based bucket coverage, new-bucket rate, and spread; map coverage to a perturbation budget and inject Gaussian noise either into the soft prompt or outputs to frustrate extraction, claiming minimal harm to benign users.

**Strengths:**

1. The paper solved a timely and important problem, and the authors provide a clear and realistic threat model.
2. Proposed attack is simple and effective at stage 1 and the learnt inversion model yields usable prompts on unseen tasks at stage 2.
3. The paper provides a concise end-to-end runtime analysis showing that the proposed two-stage attack is far faster than prompt tuning. CAP adds only moderate overhead suitable for deployment.

**Weaknesses:**

1. Limited evaluation beyond mid-scale classification: experiments focus on T5/RoBERTa/GPT-2/Pythia and classification tasks. Evidence for generation tasks (e.g., summarization) is weak or inconsistent and should be expanded with generation metrics (ROUGE/BERTScore) and CAP-on vs CAP-off comparisons.
2. Table 2 shows when trained on YELP and evaluated on MOVREV, the Adversary (CAP Off) accuracy is 89.33% (+7.2 points to target). This is intriguing, could we provide some insight for the underlying reason? Or a variance analysis with multiple seeds may help.

**Questions:**

1. Will identical inputs from the same benign user may produce slightly different behavior over time? Can we guarantee stability and user-visible variance?
2. Can CAP be fooled by coherent multi‑task users (e.g., a legitimate pipeline with naturally diverse inputs)?

---

> ### Author Response · Authors · 2025-11-21
> **Rebuttal**
>
> We thank the Reviewer for the positive and detailed feedback. We appreciate that the Reviewer finds our work important and recognizes that our effective two-stage attack outperforms prompt tuning both in usability and runtime overhead. We are happy that our threat model is clear and realistic. We provide detailed answers to the concerns and questions in-line below:
>
> Weaknesses:
> >**Table 2 shows when trained on YELP and evaluated on MOVREV, the Adversary (CAP Off) accuracy is 89.33% (+7.2 points to target). This is intriguing, could we provide some insight for the underlying reason? Or a variance analysis with multiple seeds may help.**
>
> As per your valuable suggestion, we experiment with 5 different seeds and include the results in the table below. After doing variance analysis, we find that the Adversary (CAP Off) performance is 81.81% which is less than and comparable to the target 82.10%. Therefore, overall, the results are consistent and expected. The unusually high MOVREV accuracy for one seed is a consequence of the stochastic prompt-augmentation during inversion. The inversion model learns from some augmented prompt embeddings, and different seeds produce slightly different geometric regularizations. So, this behavior where the adversary exceeds the target model is not observed across all seeds
>
> | Seed | Target (%) | Adversary(%) |
> |------|-------|------|
> | 45   | 82.10 | 80.40 |
> | 1339 | 82.10 | 89.33 |
> | 100  | 82.10  | 79.10 |
> | 1335 | 82.10 | 79.70 |
> | 50   | 82.10 | 80.52 |
>
> >**Will identical inputs from the same benign user may produce slightly different behavior over time? Can we guarantee stability and user-visible variance?**
>
> For benign users submitting identical inputs repeatedly, CAP ensures near-deterministic outputs because query-invariance assumes the soft prompt remains fixed during inference. However, CAP deliberately drifts the soft prompt based on accumulated coverage metrics, so over time, as you mentioned, even identical queries may produce slightly different outputs as coverage incrementally increases. The coverage grows quite slowly for benign users, keeping perturbation near baseline ϵ=0.0005, which results in negligible performance degradation for them.
>
> >**Can CAP be fooled by coherent multi‑task users (e.g., a legitimate pipeline with naturally diverse inputs)?**
>
> We acknowledge that sybil adversaries can indeed fool CAP and hence also provide a defense against them. We also added the section on Sybils below for your reference and also add this to paper (Section 7, Table 5):
> We consider an adversary who queries the API from $n$ accounts. For every account, the model outputs (probabilities), which are released to the adversary. The adversary collects these model outputs to train the inversion model. Sybil adversaries try to circumvent our defense by carefully partitioning their diverse queries across multiple accounts and mixing them with less diverse queries. They do this to ensure that our coverage-aware defense does not flag the user as malicious due to the low embedding coverage. Further, the sybil adversary can gather these outputs from different accounts, effectively achieving a high embedding coverage, while still obtaining minimally perturbed outputs and evading detection.
> To mitigate the risk of sybil-based attacks, we introduce a defense that perturbs model outputs using a random affine transformation. While the random affine transformation remains consistent for a given legitimate user who queries the API from a single account, sybils obtain model outputs tampered by random affine transformations for every account they query the API with. In other words, we apply different affine transformations per account. To evaluate the effectiveness of this defense, we simulate a downstream classification task by using the obtained model outputs. With the training of the classifier on consistently transformed model outputs, we observe that high utility is preserved for legitimate users, with minimal degradation. On the other hand, Sybil adversaries, who receive inconsistent model outputs transformed by random affine transformations from every account, observe a significant degradation in the utility, due to ineffective learning. Thus, we show that our defense also prevents sybil attacks by preserving the consistency of model outputs for legitimate users, while disrupting it for sybil adversaries. The results in the table below show the effectiveness of our defense against Sybil adversaries.
>
> | User Type | # Queries     | Downstream Task | Downstream Accuracy (%) |
> |-----------|---------------|------------------|---------|
> | LEGIT     | 2000          | SST2             | 95.00    |
> | SYBIL     | 500 × 4       | SST2             | 50.60   |
> | LEGIT     | 2000          | MovRev           | 88.80  |
> | SYBIL     | 500 × 4       | MovRev           | 11.00 |
> | LEGIT     | 2000          | IMDB             | 95.20 |
> | SYBIL     | 500 × 4       | IMDB             | 4.80 |

---

### Meta-Review · Area_Chair_R8yo · 2026-01-06

**Summary:**

he paper studies the vulnerability of soft-prompt-tuned LLM deployments and proposes: (1) Attack: A two-stage black-box attack, including distillation stage (learn a functionally equivalent soft prompt by minimizing KL divergence to the victim’s next-token probabilities) and inversion stage (train a small transformer to map next-token probability vectors to soft-prompt embeddings that transfer across tasks); (2) Defense: Coverage-Aware Perturbation (CAP), track query diversity via LSH-based bucket coverage, new-bucket rate, and spread; map coverage to a perturbation budget and inject Gaussian noise either into the soft prompt or outputs to frustrate extraction, claiming minimal harm to benign users.

However, almost all reviewers point that experiments are conducted under small scale models and datasets.
Thus, all reviewers gave negative scores, I believe this paper in current form is not good enough for an accept.

**Reviewer Concerns:**

Almost all reviewers point that experiments are conducted under small-scale models and datasets.
However, the authors did not provide sufficient experiments on large-scale models during the rebuttal.

**Reviewer Scores:**

No.

---

### Decision · Program_Chairs · 2026-01-26

Reject